# Furanocoumarins from *Ruta chalepensis* with Amebicide Activity

**DOI:** 10.3390/molecules26123684

**Published:** 2021-06-16

**Authors:** Aldo Fabio Bazaldúa-Rodríguez, Ramiro Quintanilla-Licea, María Julia Verde-Star, Magda Elizabeth Hernández-García, Javier Vargas-Villarreal, Jesús Norberto Garza-González

**Affiliations:** 1Laboratorio de Fitoquímica, Facultad de Ciencias Biológicas, Universidad Autónoma de Nuevo León (UANL), Av. Universidad S/N, Cd. Universitaria, San Nicolás de los Garza C.P. 66455, Nuevo León, Mexico; aldo.bazalduarg@uanl.edu.mx (A.F.B.-R.); maria.verdest@uanl.edu.mx (M.J.V.-S.); 2Laboratorio de Fisiología Celular y Molecular, Centro de Investigaciones Biomédicas del Noreste (CIBIN), Dos de Abril Esquina con San Luis Potosí, Monterrey C.P. 64720, Nuevo León, Mexico; magda.hernandezgr@uanl.edu.mx; 3Laboratorio de Biología y Fisiología Molecular y Celular, Centro de Investigaciones Biomédicas del Noreste (CIBIN), Dos de Abril Esquina con San Luis Potosí, Monterrey C.P. 64720, Nuevo León, Mexico; jvargas147@yahoo.com.mx; 4Departamento de Ciencias Básicas, Universidad de Monterrey, Av. Ignacio Morones Prieto 4500 Pte., San Pedro Garza García C.P. 66238, Nuevo León, Mexico; jesusn.garza@udem.edu

**Keywords:** parasitic diseases, amoebiasis, Mexican traditional medicine, bioguided isolation, natural products, antiprotozoal agents

## Abstract

*Entamoeba histolytica* (protozoan; family Endomoebidae) is the cause of amoebiasis, a disease related to high morbidity and mortality. Nowadays, this illness is considered a significant public health issue in developing countries. In addition, parasite resistance to conventional medicinal treatment has increased in recent years. Traditional medicine around the world represents a valuable source of alternative treatment for many parasite diseases. In a previous paper, we communicated about the antiprotozoal activity in vitro of the methanolic (MeOH) extract of *Ruta chalepensis* (Rutaceae) against *E. histolytica*. The plant is extensively employed in Mexican traditional medicine. The following workup of the MeOH extract of *R. chalepensis* afforded the furocoumarins rutamarin (**1**) and chalepin (**2**), which showed high antiprotozoal activity on *Entamoeba histolytica* trophozoites employing in vitro tests (IC_50_ values of 6.52 and 28.95 µg/mL, respectively). Therefore, we offer a full scientific report about the bioguided isolation and the amebicide activity of chalepin and rutamarin.

## 1. Introduction

Amoebiasis is the term for a parasitic infection triggered by the protozoan *Entamoeba histolytica*. This disease represents one of the most widespread parasite maladies in developing countries [1,2], making it a significant public health problem [3,4].

In Mexico, this infection is considered an endemic illness [5,6,7] and represents the most common parasitic disease found in the general population [8,9]. It is also more frequently found in the intestinal amoebiasis of Mexican children, especially newborn and school children [10].

Amoebiasis frequently produces an intestinal infection that can transform into an extra-intestinal infection throughout the portal veins, affecting the liver and producing a more significant hepatic lesion, in addition to lesions in the lungs, skin, and brain [11,12]. Following malaria, amoebiasis represents the second leading factor of death caused by parasite infections [13]. Nearly 50 million people develop acute amoebiasis, and 40,000–100,000 deaths caused by this sickness happen per year worldwide [14,15].

It is important to highlight that the chemotherapy used against *E. histolytica* has shown significant advancement. Nevertheless, the treatment’s abandonment produces chronic patients, illness propagators, and develops a multidrug resistance parasite [10,16].

The most utilized therapeutic drug for the medication of this infection is metronidazole, but, owing to the drug’s undesired side effects [17,18] and considering the burgeon of resistant strains of *E. histolytica* against it, new antiprotozoal agents are required [19,20]. Plants are a good source of natural products that have been used in the treatment of protozoa illness [21,22,23,24], and Mexican traditional medicine can offer many plants that could be useful for developing treatment against *E. histolytica* [25]. In a previous paper, we discussed the antiamoebic activity in vitro of the MeOH extract of *Ruta chalepensis* [26]. *Ruta chalepensis* is a Mediterranean plant introduced in Mexico and used by different ethnic groups to treat gastrointestinal illnesses such as stomachache, diarrhea, dysentery, and nausea [27]. Additionally, there are reports of emmenagogue, anthelmintic, antirheumatic, antihypertensive, abortive, and anti-inflammatory activity for this plant [28,29]. *R. chalepensis* possess many known furanocoumarins, coumarins, alkaloids, quinoline alkaloids, and flavonoids distributed in leaves, stems, and roots [28,30,31].

This research aims to isolate and identify substances responsible for the amebicide activity of *R. Chalepensis*.

## 2. Results

### 2.1. Bioguided Isolation of Furanocoumarins from Ruta chalepensis

The following diagram (Figure 1) shows the number of chromatography columns implemented (silica gel or Sephadex), as well as the percentage of inhibition against *E. histolytica* of the most relevant fractions during the bioguided fractionation of the MeOH extract of *R. chalepensis* to obtain the furanocoumarins Rutamarin (**1**) and Chalepin (**2**) in high purity (TLC; see Appendix A).

Both compounds were characterized by different spectroscopic techniques.

The unambiguous assignment of the ^13^C-NMR spectrum of the isolated furanocoumarins was obtained from ^1^H-^1^H COSY, NOESY, HSQC, and HMBC spectra (see Appendix A).

Rutamarin (**1**) was recovered as a white amorphous solid (m. p. 104 °C; Lit. 107–109 °C; [32]. The spectroscopy data of this compound corresponded to a previous compound reported as rutamarin, **1** [32].

Chalepin (**2**) was obtained as a yellow amorphous solid (m. p. 125–126 °C; Lit. 117–118 °C) [33]. The spectroscopy data of this compound corresponded to a previous compound reported as chalepin, **2** [34].

Both compounds are structurally related to chalepensin (**3**), a furanocoumarin also occurring in this plant [26]. See Figure 2.

### 2.2. Determination of IC_50_ for Rutamarin (***1***) and Chalepin (***2***)

The in vitro assay for the isolated furanocoumarins against trophozoites of *E. histolytica* showed significant activity.

Figure 3 and Figure 4 show the 50% inhibitory concentration of each compound calculated using a probit analysis, considering a 95% confidence level.

## 3. Discussion

*Ruta chalepensis* is a medicinal plant used worldwide for its wide range of medicinal purposes [35]. However, the antiparasitic activity against *Entamoeba histolytica* of this plant has not been sufficiently disclosed in ethnobotanical reports until now. Antiparasitic activity of *R. chalepensis* has been previously evaluated on helminths of veterinary importance [36,37] and protozoa of clinical importance such as *Leishmania infantum*, *L. major* [38] and *Giardia lamblia* [39,40]. The antiparasitic activity over *E. histolytica* by the methanolic crude extract was previously reported [26], showing moderate activity with an IC_50_ of 60.07 μg/mL. The antiparasitic activity of the furanocoumarin chalepensin (**3**), isolated from the hexane partition of *R. chalepensis* by Quintanilla-Licea et al. [26], was also moderate, showing an IC_50_ of 45.95 μg/mL. The antiamoebic activity of rutamarin (**1**) and chalepin (**2**) isolated in this research work is slightly higher (IC_50_ of 6.52 and 28.95 μg/mL, respectively) but far less effective than metronidazole (IC_50_ 0.205 µg/mL) [26]. We can therefore consider the three furanocoumarins as responsible for the antiparasitic activity of *R. chalepensis*. Furanocoumarins are part of the chemical components with the most significant presence in *R. chalepensis* [41]. Rutamarin (**1**) and chalepin (**2**) have been previously isolated from this plant [42]; our research group briefly described the amebicide activity of these furanocoumarins at a scientific congress [43,44]. We are now offering a full scientific report about the bioguided isolation and identification of these amebicide compounds. These results may support the use of *R. chalepensis* as an alternative treatment for amoebiasis in traditional Mexican medicine, as it is for other plants of equal importance [45,46]. The toxic activity of coumarins on eukaryotic cells has received an important analysis suggesting a partial mechanism of action for compounds of this nature where cell injury is caused by damage to DNA [47]. Chalepin (**2**) has shown specific toxicity on normal cells and cancer cell lines, for example HT29 human colon carcinoma cells with cytotoxic activity by 55.1 μM. Although rutamarin (**1**), in some analyses, showed selectivity with its cytotoxic activity at a concentration of 1.12 μM, it was also reported that it generates significant alterations in the cell growth of BCBL-1 cells at a concentration of 5.60 μM [48,49]. A certain toxicity on normal mammalian cells at the respective concentrations of the IC_50_ of compounds **1** and **2** (18.29 and 92.08 μM, respectively) could be expected. The furanocoumarins isolated from *R. chalepensis* (**1-3**) have outstanding antiamoebic activity, but the presence of an acetyl group in rutmarin may increase the antiparasitic activity, producing a better complex with DNA and therefore causing more considerable cellular damage [47].

## 4. Materials and Methods

### 4.1. General Experimental Procedures

NMR spectra were acquired on an Avance DPX 400 Spectrometer (Bruker, Billerica, MA, USA) working at 400.13 MHz for ^1^H and 100.61 MHz for ^13^C. Melting points were measured on Electrothermal 9100 equipment (Electrothermal Engineering Ltd., Southend-on-Sea, UK).

Thin-layer chromatography (TLC) was developed on pre-coated silica gel plates (Merck, Kenilworth, NJ, USA Silica Gel 60 F254). Visualization of the components of the crude extract or pure compounds was made using UV light. Open column chromatography was implemented with silica gel of a 60–200 mesh (J. T. Baker, Phillipsburg, NJ, USA) and Sephadex LH-20 (Sigma-Aldrich, St. Louis, MO, USA, LH20100-500G).

### 4.2. Plant Material

*R. chalepensis, L. Rutaceae* was collected near the city of Aramberri, Nuevo León State in northern Mexico, 24°19′13″ N, 99°54′55″ W. It was dried using a light chamber at 38 °C and was then powdered with a manual grain. Plant material was placed at the herbarium of the Facultad de Ciencias Biológicas in the Universidad Autónoma de Nuevo León, with a register number 025579.

### 4.3. Drugs and Reagents

The bioassay was made in the PEHPS medium, provided by the Centro de Investigación Biomédica del Noreste, IMSS [50]. CTR^®^ Scientific (Mexico) purchased organic solvents: *n*-hexane, chloroform (CHCl_3_), Ethyl acetate (EtOAc), Methanol (MeOH), and dimethyl sulfoxide (DMSO).

### 4.4. Extraction of Plant Material from R. chalepensis

Six hundred grams of leaves and stems from *R. chalepensis* were dried and powdered, separated in packets of 60 g, and subjected to extraction with 600 mL of MeOH, each using Soxhlet equipment for 40 h. The MeOH was eliminated in a rotary evaporator. The crude extract was preserved at 4 °C until it was needed.

### 4.5. Parasite and In Vitro Amebicide Assay

#### 4.5.1. Microorganisms

The trophozoites of *Entamoeba histolytica* (strain HM-1:IMSS) were acquired from the Centro de Investigación Biomédica del Noreste (CIBIN) microorganism culture collection in Monterrey (Mexico). The parasites were grown axenically and kept up in peptone combined with pancreas extract, liver extract and bovine serum (PEHPS medium, so designated based on the Spanish initials of its major components). The microorganisms were utilized at the log phase of growth (2 × 10^4^ cells/mL) by all of the tests carried out [51,52].

Next, 2 × 10^5^ trophozoites of *E. histolytica* in 5 mL of the PEHPS medium (with the 10% bovine serum added) were inoculated in 13 mm × 100 mm screw cap test tubes and incubated at 36.5 °C for 120 h to establish the growth curve for *E. histolytica*. Every 24 h, the number of trophozoites was evaluated to determine the medium’s growth parameters [53]. The procedure was performed in 3 separate assays per triplicate.

#### 4.5.2. In Vitro Test for *Entamoeba histolytica*

Each sample was dissolved in the DMSO and standardized to 150 µg/mL by adding a mixture of *E. histolytica* trophozoites at a logarithmic phase in the PEHPS medium with the 10% bovine serum. Afterward, vials were incubated for 72 h and then introduced into iced water for 20 min. The number of dead trophozoites per milliliter was estimated by using a hemocytometer. Each assay was implemented in triplicate. The positive control was metronidazole. The negative control was an *E. histolytica* suspension in the PEHPS medium with no extract added. The percentage of inhibition was determined by evaluating the number of dead trophozoites in the samples and the negative controls [53].

#### 4.5.3. In Vitro IC_50_ Evaluation

Each sample was dissolved in the DMSO and standardized to 150, 75, 37.5, 18.75, and 9.375 µg/mL by adding a mixture of *E. histolytica* trophozoites at a logarithmic phase in the PEHPS medium with the 10% bovine serum. Afterward, vials were incubated for 72 h and then introduced into iced water for 20 min. The number of dead trophozoites per milliliter was estimated by using a hemocytometer. Each assay was implemented in triplicate. The positive control was Metronidazole. The negative control was an *E. histolytica* suspension in the PEHPS medium with no extract added. The percentage of inhibition was determined by evaluating the number of dead trophozoites in the samples and the negative controls. The 50% inhibitory concentration of each sample was determined using a probit analysis, taking into account a 95% confidence level [53].

### 4.6. Bioassay-guided Fractionation

The crude MeOH extract of *R. chalepensis* (124 g) with an amebicide activity of 90.50% at 150 µg/mL [26] was dissolved again in one L methanol and distributed into four portions of 250 mL, each to make a liquid–liquid partition with *n-*hexane (750 mL each portion). The hexane partition was evaporated under vacuum. Column chromatography on the silica gel of this hexane residuum led to the isolation of chalepensin (**3**), Figure 1 [26]. The MeOH phase was then concentrated under vacuum until obtaining 50 mL and was added drop by drop into 200 mL of distilled water under continuous agitation. The resulting suspension was put through a liquid–liquid partition with EtOAc (750 mL). After EtOAc evaporation, 20.0 g of a resin with a good amebicide activity (84.82% of growth inhibition at 150 µg/mL) was obtained. The EtOAc partition was distributed in ten portions of 2 g; each one were subjected to open column chromatography (30 cm × 2 cm) on the silica gel (25 g each), eluting with stepwise gradients of CHCl_3_-EtOAc (100:0, 90:10, 80:20, 70:30, 60:40, 50:50, 40:60, 30:70, 20:80, 10:90, 0:100 *v/v*, each at 50 mL) and EtOAc-MeOH (100:0, 90:10, 80:20, 70:30, 60:40, 50:50 *v/v*, each at 50 mL). All fractions obtained were gathered and pooled based on their TLC (CHCl_3_-EtOAc; 9.5:0.5) profile to yield nine fractions (A1-A9). The in vitro amebicide assay exhibited that A3 and A4 had the highest growth of inhibition (95.84% and 94.18%, respectively). The fraction A3 (1.55 g) was subjected to open column chromatography on the silica gel, eluting with gradients of CHCl_3_-EtOAc (100:0, 90:10, 80:20, 70:30, 60:40, 50:50, 40.60 *v/v*; each at 50 mL). The fractions were gathered and pooled based on their TLC (CHCl_3_-EtOAc; 9.5:0.5) profile in five fractions (B1–B5). The fraction B2 (716 mg) with the best amebicide activity (an 81.75% growth inhibition) was submitted to open column chromatography on the silica gel, eluting with gradients of CHCl_3_-EtOAc (100:0, 90:10, 80:20, 70:30, 60:40, 50:50, 40.60, 30:70, 20:80, 10:90, 0:100 *v/v*; each at 50 mL). The fractions obtained were collected and pooled on their TLC (CHCl_3_-EtOAc; 9.5:0.5) profile in four fractions (C1–C4). The fraction C2 (540 mg, a 92.01% growth inhibition) was submitted to open column chromatography on the silica gel, eluting with gradients of CHCl_3_-EtOAc (100:0, 99:01, 95:05, 90:10, 85:15, 80:20, 70:30, 60:40 *v/v*; each at 50 mL). The fractions obtained were collected and pooled on their TLC (CHCl_3_-EtOAc; 9.5:0.5) profile in three fractions (D1–D3). The fraction D2 (499 mg, an 86.72% growth inhibition) was subjected to crystallization in methanol, obtaining a white amorphous solid (365 mg, an 88.56% growth inhibition), which was submitted to open column chromatography on Sephadex LH-20, eluting with MeOH. The fractions obtained were gathered and pooled on their TLC (CHCl_3_-EtOAc; 9.5:0.5) profile in five fractions (E1–E5). Fraction E2 afforded 252 mg pure rutamarin (**1**), Figure 2, with a 90.03% growth inhibition. IC_50_ = 6.52 µg/mL (18.29 µM). *R_f_* = 0.66 (CHCl_3_:EtOAc, 9.5:0.5).

The fraction A4 (476 mg) with an 94.18% of amebicide activity was submitted to open column chromatography on the silica gel, eluting with stepwise gradients of CHCl_3_-EtOAc (100:0, 90:10, 80:20, 70:30, 60:40, 50:50, 40:60, *v/v*, each at 50 mL). The obtained fractions were collected and pooled using the information obtained from their TLC profile in six fractions (F1–F6). The fraction F4 (193 mg, an 89.43% growth inhibition) was subjected to open column chromatography on Sephadex LH-20, using methanol as an eluent. The fractions obtained were collected and pooled on their TLC (CHCl_3_-EtOAc; 9:1) profile into four fractions (G1–G4). The fraction G2 (81 mg, a 90.41% growth inhibition) was submitted to open column chromatography on Sephadex LH-20, using methanol as an eluent. The fractions obtained were collected and pooled on their TLC (CHCl_3_-EtOAc; 9.5:0.5) profile in four fractions (H1-H3). Fraction H2 afforded 66 mg pure chalepin (**2**), Figure 3, with a 94.50% growth inhibition. IC_50_ = 28.95 µg/mL (92.08 µM). ***R_f_* =** 0.38 (CHCl_3_:EtOAc, 9:1).

## 5. Conclusions

This study describes the bioguided isolation [54] of two amebicide constituents of *R. chalepensis.* The furanocoumarins rutamarin (**1)** and chalepin (**2**) represent the most active chemical components of *R. chalepensis* against the protozoa *E. histolytica* with IC_50_ values of 6.52 and 28.95 μg/mL, respectively. To the best of our knowledge, this is the first report about the amebicide activity of chalepin and rutamarin. Even though the amebicide activity of both compounds are far less effective than metronidazole (IC_50_ 0.205 µg/mL), the IC_50_ values obtained in our research for rutamarin and chalepin may also be used as the basis for incorporating *Ruta chalepensis* extracts into conventional and complementary medicine for the therapy of amoebiasis and other infectious diseases.

## Figures and Tables

**Figure 1 molecules-26-03684-f001:**
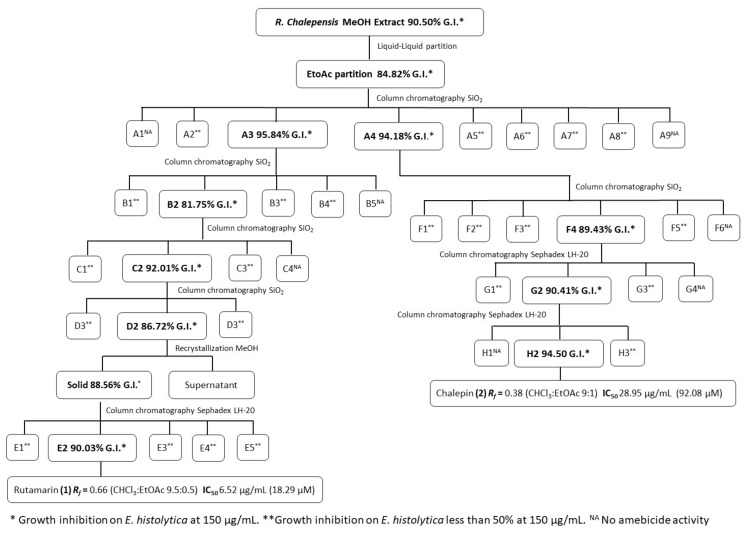
General scheme for the bioguided isolation of compounds with antiamoebic activity from *Ruta chalepensis*.

**Figure 2 molecules-26-03684-f002:**
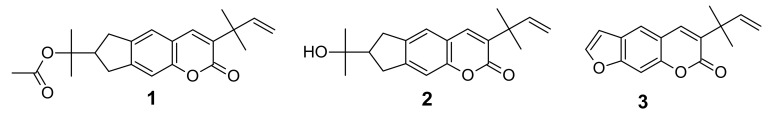
Structure of furanocoumarins isolated from *Ruta chalepensis* with amebicide activity.

**Figure 3 molecules-26-03684-f003:**
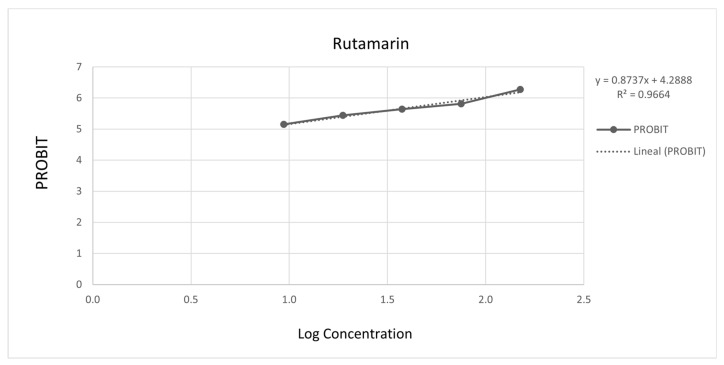
Antiprotozoal activity of Rutamarin **1** against *Entamoeba histolytica*. 90.03% Growth inhibition at 150 µg/mL, IC_50_ = 6.52 µg/mL (18.29 µM).

**Figure 4 molecules-26-03684-f004:**
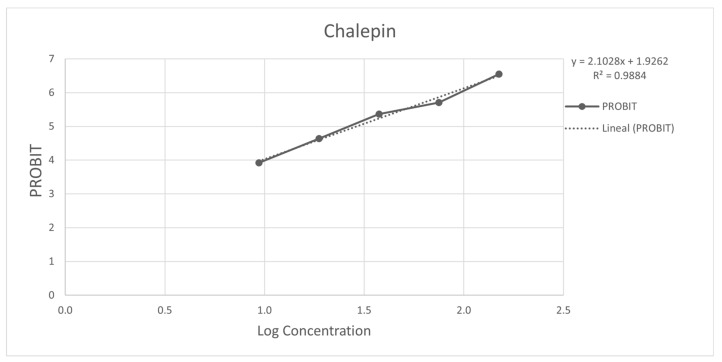
Antiprotozoal activity of Chalepin **2** against *Entamoeba histolytica* 94.50% Growth inhibition at 150 µg/mL, IC_50_ = 28.95 µg/mL (92.08 µM).

## Data Availability

Not applicable.

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
