# Peer review of "Furanocoumarins from Ruta chalepensis with Amebicide Activity"

_molecules, 2021, doi:10.3390/molecules26123684_

Round 1

Reviewer 1 Report

The present manuscript reports on the bioguided isolation of two amebicidal furanocoumarines from Ruta halepensis. The methods applied are up-to-date and appropriate, the conclusions are supported by the results and the results are valuable. However a couple of points need additional attention by the Authors:

  1. “Results” section: The NMR data of known compounds belong to the Supplementary material, not to the Results
  2. . Again in “Results”, a brief description of the bio-guided isolation must be presented, to demonstrate how the activity was used for guidance.

Author Response

Reviewer 1-Authors

"Results" section: The NMR data of known compounds belong to the Supplementary material, not to the Results.

The NMR data of the identified compounds have been changed to the supplementary material section (Fig. 37)

Again in "Results", a brief description of the bio-guided isolation must be presented to demonstrate how the activity was used for guidance.

Section 2.1 was renamed Bioguided isolation of furanocoumarins from Ruta chalepensis, and a figure with the following legend was added.

Figure 1. General scheme for the bioguided isolation of compounds with antiamoebic activity from Ruta chalepensis.

Reviewer 2 Report

The work is interesting but it needs to be improved.

  • Figures 2 and 3: what's the y axis ? growth inhibition ? please specify. moreover, add the data of negative and positive controls in the plot. You used triplicate, maybe you should add also SD value 
  • 4.5.3. Line 185: Does the negative control include also DMSO in the medium ? You must verify the potential toxicity of max concentration. 
  • In order to verify the amebicide activity, you should use also a viability assay like a colorimetric assay (MTT assay, ...)
  • You could do a comparison with metronidazole

Author Response

Reviewer 2-Authors

Figures 2 and 3: what's the y axis? Growth inhibition? Please specify. Moreover, add the data of negative and positive controls in the plot. You used triplicate, maybe you should add also SD value

In Figures 2 and 3 (3 and 4 in this version), the "Y" axis corresponds to the PROBIT value, which is related to the percentage of mortality according to Finney's method published in 1971 and described in: Arambašic, M.B, and Randhawa, M.C. 2014. Comparison of the methods of Finney and Miller-Tainter for the calculation of LD50 values. World Applied Sciences Journal 32, 2167-2170.

DOI: 10.5829/idosi.wasj.2014.32.10.9132

Regarding the observation of adding the standard deviation and negative control, we confirm that a second graph of sigmoidal dose response was already added, which is found in the supplementary material (Figs. 38 and 39).

4.5.3. Line 185: Does the negative control include also DMSO in the medium? You must verify the potential toxicity of max concentration. 

The methodology used to develop the bioassay is based on Mata-Cárdenas, B.D., Vargas-Villareal, J., Said-Fernández, S. 2008. A new vial microassay to screen antiprotozoal drugs. Pharmacologynline 1, 529-537. Following this methodology, the negative control is composed only of the culture medium and the trophozoites of E. histolytica. However, the highest concentration of DMSO to which the parasitic is exposed is 0.25%, a percentage that does not represent toxicity to this protozoan (Naidu, R., et al., 2018. A reference document on Permissible Limits for solvents and buffers during in vitro antimalarial screening. Scientific Reports 8, 14974. DOI:10.1038/s41598-018-33226-z; Cevallos, A.M., et al. 2017. Differential effects of two widely used solvents, DMSO and ethanol, on the growth and recovery of Trypanosoma cruzi epimastigotes in culture. Korean J. Parasit. 55, 81-84. https://doi.org/10.3347/kjp.2017.55.1.81)

In order to verify the amebicide activity, you should use also a viability assay like a colorimetric assay (MTT assay, ...)

The methodology used to develop the bioassay is based on Mata-Cárdenas, B.D., Vargas-Villareal, J., Said-Fernández, S. 2008. A new vial microassay to screen antiprotozoal drugs. Pharmacologyonline 1, 529-537.

This methodology does not use a colorimetric method to evidence biological activity since it is not necessary. In Neubauer's chamber, only live trophozoites and cellular remains are observed.

You could do a comparison with metronidazole

A comparison between the IC50 values of compounds 1 and 2 versus the IC50 value of metronidazole was added in the discussion section, lines 154-155.

Reviewer 3 Report

See attached PDF

Author Response

Reviewer 3-Authors

  1. The fractionation of the compounds indicates that they were "pure" at conclusion of the efforts. How was this resolved? Details of the experiments used to determine this should be included.

In the section "General Experimental Procedures" it is described that the control of the fractionation and purification of the isolated compounds was performed using thin-layer chromatography. The supplementary material includes chromatographic evidence and the Rf of compounds 1 and 2 in Figure 40.

  1. Sigmoidal dose response curves should be included (either in lieu of or in addition to, the PROBIT analysis). They allow simple visual inspection of the nature of the inhibition, and the quality of the assay.

We confirm that it was added graphs of sigmoidal dose response, which is found in the supplementary material (Figs. 38 and 39).

  1. The discussion, Lines 125-128, mentions toxicity of the compounds to human cells. What are the dose ranges required for toxicity? This will inform readers of the potential value of the compounds vs. amoebae – whether there is a sufficiently large selective index. Even if not, those numbers are informative for the same reason and do not dampen enthusiasm for publication. Additionally, the idea that rutamarin does not damage normal cells may be true at some concentrations – but when does it become toxic? That info (from papers 48,49) should also be included.

We welcome that point. The publication was supplemented with the cytotoxicity data requested for chalepin and rutamarin. View lines 166-173

  1. The last sentence of the discussion is too speculative here, as the SAR is insufficient against the amoebae to identify the importance of the acetyl group. It is ok to say "may increase".

We accepted the suggestion, and the proposed change was made in lines 173-175

  1. The live/dead assay details are unclear. How are dead cells distinguished from those that are not moving? If they are not obliterated, this could be very challenging in the absence of a vital dye (PI exclusion, or a CellTitre reagent, for example). These values are important to the findings in the paper, so some sort of standard assay should be used.

The methodology used to develop the bioassay is based on Mata-Cárdenas, B.D., Vargas-Villareal, J., Said-Fernández, S. 2008. A new vial microassay to screen antiprotozoal drugs. Pharmacologyonline 1, 529-537.

This methodology does not use a colorimetric method to evidence biological activity since it is not necessary. In Neubauer's chamber, only live trophozoites and cellular remains are observed.

  1. The paper is at times difficult to read due to grammar issues, which dampens overall enthusiasm (examples are included below, but are by no means exhaustive):
  2. Abstract, line 19 – "least developed countries" (unclear meaning)
  3. "first", line 37 – not clear – "represents the most common parasitic disease…?
  4. Line 43, "…factor of decease provoked by parasite infections" – unclear
  5. "demises" (line 45) should be "deaths"

Drafting changes were made to the marked sections.

Reviewer 4 Report

In the present study, Rodríguez et al. confirmed a previously described anti-Entamoeba histolytica activity of a MeOH extract from R. chalepensis and they now found that two furocoumarins purified from these extracts, namely rutamarin and chalepin, exhibit considerable cytotoxic activity against in vitro cultivated E. histolytica trophozoites. To a certain extent, the authors describe the procedure how chromatographic purification and subsequent identification of these two compounds was achieved. Unfortunately, however, many details of the respective experiments are not at all, or at least incompletely, documented in the present paper. In particular, a follow-up documentation of the bio-guided isolation of the two compounds (HPLC-based analysis and growth inhibition testing of raw extract as well as of the different chromatographic fractions) is missing (should be added as supplementary data). Furthermore, E. histolytica growth curves demonstrating the inhibitory concentrations of the extract/compounds should be provided in a supplementary file. Here, respective data from the positive (metronidazole treatment) and negative control experiment should also be included. Additionally, IC50 values should be given as µM (or nM) instead of µg/ml. Finally, I recommend that the authors should determine the IC50 values of the two compounds (plus metronidazole) in a mammalian cell culture (e.g using BCBL-1 cells or a hepatic cell line). Previous findings indicated that, at least at low concentrations, rutamarin does not have a significant cytotoxic activity on mammalian cell lines. However, I wonder whether those concentrations cytotoxic for E. histolytica have also an inhibitory effect on in vitro growth of mammalian cells. This additional experiment would at least give a preliminary idea about the selective toxicity of rutamarin (and possibly also chalepin) in terms of its later application in an anti-E. histolytica chemotherapy.

Author Response

Reviewer 4-Authors

Unfortunately, however, many details of the respective experiments are not at all, or at least incompletely, documented in the present paper.

In particular, a follow-up documentation of the bio-guided isolation of the two compounds (HPLC-based analysis and growth inhibition testing of raw extract as well as of the different chromatographic fractions) is missing (should be added as supplementary data).

In the section "General Experimental Procedures" it is described that the control of the fractionation and purification of the isolated compounds was performed using thin layer chromatography. The supplementary material includes chromatographic evidence and the Rf of compounds 1 and 2 in Figure 40.

Furthermore, E. histolytica growth curves demonstrating the inhibitory concentrations of the extract/compounds should be provided in a supplementary file. Here, respective data from the positive (metronidazole treatment) and negative control experiment should also be included. Additionally, IC50 values should be given as µM (or nM) instead of µg/ml.

We confirm that it was added graphs of sigmoidal dose response, which is found in the supplementary material (Figs. 38 and 39).

Changes were made to the values from IC50  to μM in the text.

Finally, I recommend that the authors should determine the IC50 values of the two compounds (plus metronidazole) in a mammalian cell culture (e.g using BCBL-1 cells or a hepatic cell line). Previous findings indicated that, at least at low concentrations, rutamarin does not have a significant cytotoxic activity on mammalian cell lines. However, I wonder whether those concentrations cytotoxic for E. histolytica have also an inhibitory effect on in vitro growth of mammalian cells. This additional experiment would at least give a preliminary idea about the selective toxicity of rutamarin (and possibly also chalepin) in terms of its later application in an anti-E. histolytica chemotherapy

We appreciate this observation and especially the proposal to complement the study by evaluating the compounds rutamarin, chalepin, and metronidazole on mammalian cells. However, it is not possible for our working group to carry out this type of assay since we do not have human cell lines or the necessary infrastructure. On the other hand, the publication will be complemented with specific toxicity data for both compounds from scientific literature, allowing a clearer picture of the value that would have rutamarin and chalepin as amebicides. View lines 166-173.

Round 2

Reviewer 2 Report

No comments.

Reviewer 3 Report

Review of resubmission of Bazaldúa-Rodriguez et al.,  “Furanocoumarins from Ruta chalepensis with amebicide activity

This resubmission has partially addressed my concerns and the modifications that have been provided improve the work.  

Comments:

The inclusion of % growth inhibition data has improved the paper, but raises an issue with the PROBIT analysis. How can an IC50 value be accurately predicted when none of the drug concentrations tested had values <50%, which is the case for rutamarin.  An extrapolated value could have a great deal of error.

Additionally, the live/dead assay details remain murky.  The referenced paper provides insufficient details for me to assess the results here.  I am not reviewing that paper, but lack of details about the robustness of the assay (what is the Z’ score for the new approach? How does it directly compare to a simple Trypan blue exclusion assay?) make me hesitant to accept it as a valid approach. Distinguishing live cells from dead ones is essential here

Reviewer 4 Report

In the revised version of the manuscript the authors have adequately addressed my points of criticism